# Continued selection on cryptic SARS-CoV-2 observed in Missouri wastewater

**Devon A. Gregory[1], Clayton Rushford[1], Torin Hunter[1], Chung-Ho Lin[2], Christie Darby[3], Nicole Niehues[3], Elizabeth Semkiw[3], Melissa Reynolds[3], Jeff Wenzel[3], Marc C. Johnson** [1]*

**1** Department of Molecular Microbiology and Immunology, University of Missouri-School of Medicine, Columbia, Missouri, United States of America, **2** Center of Agroforestry, School of Natural Resources, University of Missouri, Columbia, Missouri, United States of America, **3** Bureau of Environmental Epidemiology, Division of Community and Public Health, Missouri Department of Health and Senior Services, Jefferson City, Missouri, United States of America

* marcjohnson@missouri.edu

**Data Availability Statement:** Raw sequences are available in NCBI's Sequence Read Archive (https://www.ncbi.nlm.nih.gov/sra) under the bioproject PRJNA748354.

## Abstract

Deep sequencing of wastewater to detect SARS-CoV-2 has been used during the COVID-19 pandemic to monitor viral variants as they appear and circulate in communities. SARS-CoV-2 lineages of an unknown source that have not been detected in clinical samples, referred to as cryptic lineages, are sometimes repeatedly detected from specific locations. We have continued to detect one such lineage previously seen in a Missouri site. This cryptic lineage has continued to evolve, indicating continued selective pressure similar to that observed in Omicron lineages.

## Author summary

Monitoring wastewater for SARS-CoV-2 has been an important part of understanding the dynamics of the virus's spread and persistence within and across communities during the pandemic. We and others have also observed variants appearing in wastewater that do not appear in clinical sampling. Many of these variants not only possess genomic changes identical to or at the same position as those that have been observed in variants of concern, particularly currently circulating Omicron variants, but often acquire the changes before they have been observed in clinical samples. We report here the continued observation of a variant in Missouri wastewater, but not in clinical sampling, that has continued to evolve, gaining genomic changes that often are the same and predate changes seen in clinical samples. These observations add to our understanding of the selective pressures driving the evolution of SARS-CoV-2.

## Introduction

Surveillance of wastewater for SARS-CoV-2 has been used to detect and track community circulating variants [1,2]. In addition to the variants of concern (VOCs) and other common

**Funding:** Work in this publication was supported by the National Institute on Drug Abuse (NIDA) of the National Institutes of Health (NIH) under award number U01DA053893. The funders had no role in study design, data collection and analysis, decision to publish, or preparation of the manuscript.

**Competing interests:** The authors have declared that no competing interests exist.

variants, wastewater surveillance has also detected variants that have not been otherwise observed [3,4]. These novel variants, which we call cryptic lineages, often persist in a sewershed for months or years and show signs of continued positive selection. The specific sources of cryptic lineages are unknown, though recent efforts have provided evidence of a human source [5]. Sequencing of immunocompromised individuals with persistent infection has also detected sequences with some similarities to the cryptic lineages [6]. However, a non-human source for some of the cryptic lineages observed in wastewater cannot be ruled out. We have previously reported on a cryptic lineage found in a Missouri metropolitan area (hence referred to as MO45) in June of 2021 [4]. Since the initial observation of this cryptic lineage, it has been sporadically detected with evolving genotypes.

## Results and discussion

We use next generation sequencing of SARS-CoV-2's receptor binding domain (RBD) to monitor variants present in Missouri wastewater. Monitoring of MO45 began in March 2021 and continues to the present with roughly weekly sampling (Fig 1). Initially this sewershed was observed to primarily have the Alpha variant with some ancestral sequences. Beta, Gamma, Delta and Mu/Theta sequences were all observed later with Delta becoming the only variant detected by August 2021. Delta was then rapidly replaced by Omicron in December 2021. Since, various Omicron variants have circulated, generally with newer variants displacing older ones, resulting in a mixture of variants co-circulating in late 2022.

In addition to the defined variants, a cryptic lineage has also been sporadically detected, first in June 2021 and last in October 2022 (Fig 2). Initial sequences of this variant had K417T T478K E484A Q493K S494P Q498H amino acid changes relative to the ancestral sequence, with E484A and Q493K only appearing in one of the two first detections. E484A and Q493K were both observed in all subsequent sequences of this cryptic lineage, while S494P was not observed again. K417T and T478K had previously been observed in the Gamma and Delta variants respectively, but the other mutations had not yet appeared in any major VOCs.

Several amino acid changes occurred subsequent to the initial observation and appeared to become fixed in the lineage. On February 2, 2022 N460K was first observed in the cryptic lineage and was thereafter fixed. Likewise, S477N and F486V were first observed in the cryptic lineage on April 5, 2022, and N440K on April 26, 2022, and in all detections since. N450D was first observed in the cryptic lineage on May 24, 2022. Though the lineage had N450Y on June 16, 2022 instead, the two subsequent detections of the lineage had N450D again. Several other changes were observed in the lineage over time, though none could be concluded to have

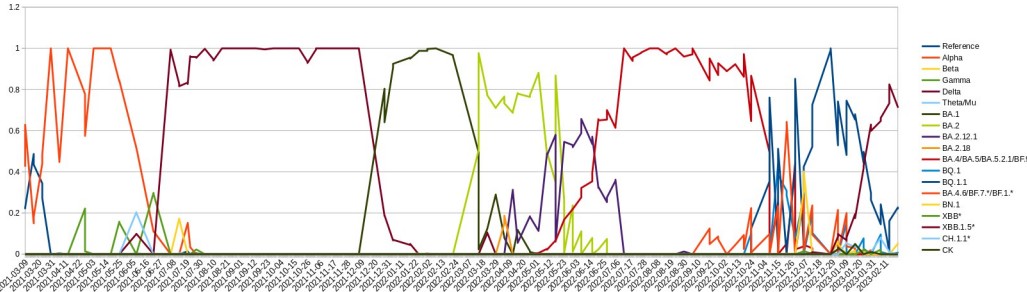

**Fig 1. Presence of Variants Over Time.** Plot of relative defined SARS-CoV-2 variant abundance detected in MO45 sewershed. Wastewater samples were processed for viral RNA extraction and SARS-CoV-2 RBD sequences were amplified and then deep sequenced. Genotype and number of reads were used to estimate the relative abundance of defined variants present in the sewershed.

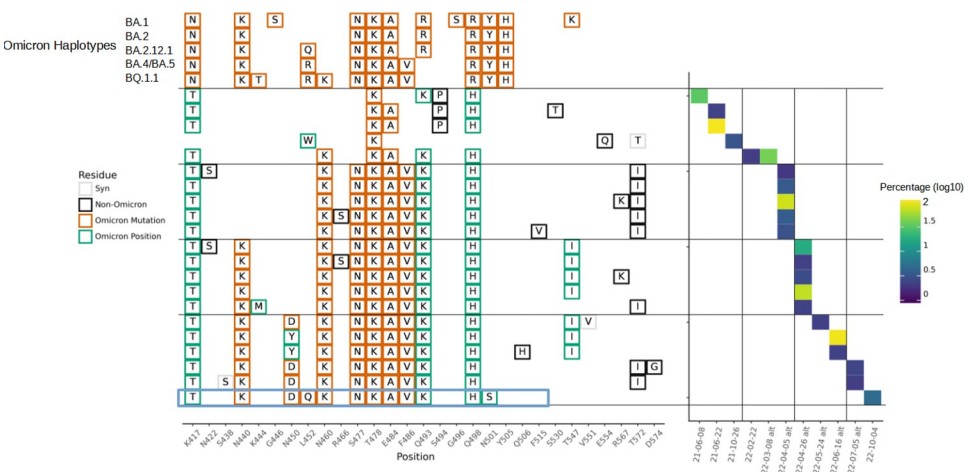

**Fig 2. Cryptic Haplotypes Detected in MO45.** Cryptic haplotypes are shown with the dates of their detection and relative abundance in the sample amplification. Select Omicron haplotypes are shown for comparison. Detection dates followed by 'alt' indicate the amplification was carried out with primers designed to exclude Omicron lineages for better detection of cryptic sequences. The boxed haplotype indicates the changes applied to Spike for neutralization assays shown in Fig 3.

become fixed. Of note, T547I and T572I both were observed in 3 samples each, but were not observed in the most recent detection. At the last detection of the MO45 cryptic lineage in October 2022, the lineage appeared remarkably similar to an Omicron lineage with 12 amino acid changes in its RBD that were all identical to, or at the same position as, changes found in Omicron lineages. It cannot be known for certain why the MO45 lineage was not detected again after October 2022. We cannot rule out that the lineage simply became undetectable by our amplification scheme, though we routinely amplified different sections of the SARS-CoV-2 genome and have not detected any outlier sequences from this sewershed.

These changes observed in the MO45 sequence are at positions that have been reported to individually contribute to immune evasion [7,8]. We wished to see the combined effects of these changes on antibody escape (Fig 3). The changes observed from October 4, 2022 were cloned into a Spike vector with the D614G change relative to Wuhan 1, and the effect of antibody neutralization in a pseudotype infectivity assay was assessed. Virus pseudotyped with D614G Spike was effectively neutralized in a dose dependent manner by LY-CoV-555, LY-CoV-016, REGN10933 and REGN10987. The Spike with the changes observed in MO45 was however almost entirely resistant to neutralization by all four antibodies.

Most of the residue changes observed in the cryptic lineage predate the changes observed in Omicron. The initial detections of the cryptic lineage, months before the emergence of Omicron, already had two changes that were to be seen ubiquitously in Omicron lineages, T478K and E484A, and three changes at the same residues as changes common in Omicron lineages, K417T, Q493K and Q498H. Likewise, N460K, which appeared in the cryptic lineage in February of 2022, did not become prevalent in an Omicron background until six months later.

The convergence of the cryptic lineage and Omicron variants suggest similar selection pressures. The origin of Omicron and the origin of the MO45 cryptic lineage are unknown. At least in some cases, cryptic lineages appear to be derived from individuals with persistent SARS-CoV-2 infections. However, as the MO45 cryptic lineage hasn't been traced, a non-human source cannot be ruled out. Since the cryptic lineage in some cases acquired changes prior to Omicron, continued monitoring of waste water for such cryptic lineages may provide foreknowledge of changes, or at least the position of changes, likely to be selected for in the circulating Omicron variants.

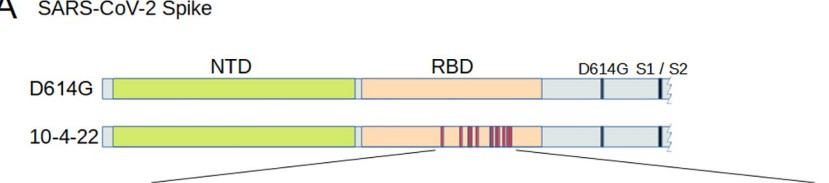

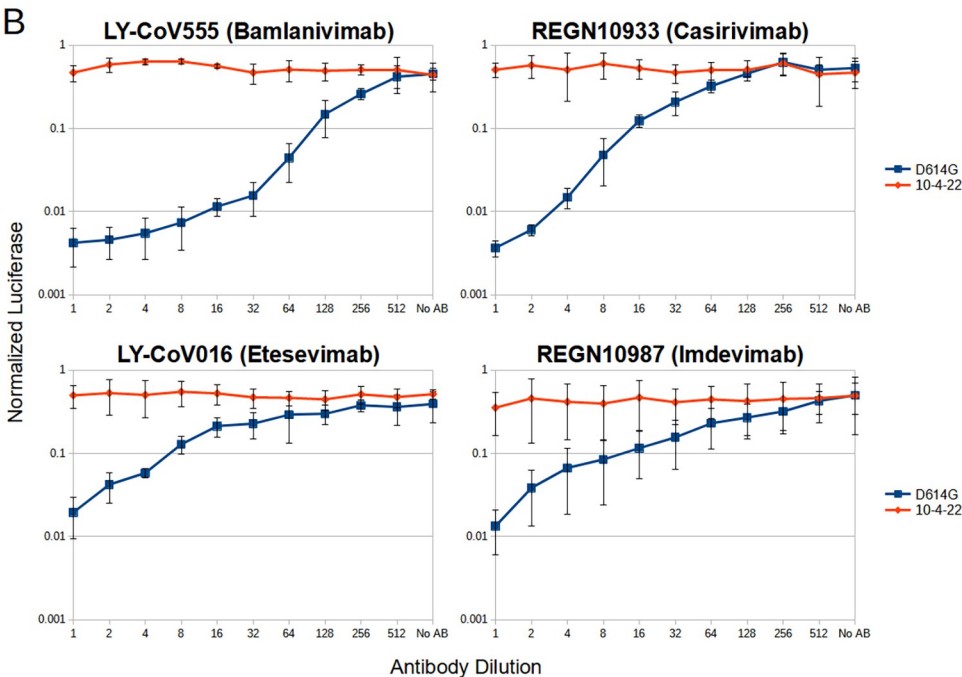

**Fig 3. Neutralizing antibody activity against MO45 cryptic lineage.** A. Schematics of SARS-CoV-2 Spike S1 with residue changes relative to Wuhan 1 used to assess antibody neutralization. B. HIV virions that express *Gaussian* luciferase upon infection were pseudotyped with either SARS-CoV-2 Spike that had only the D614G polymorphism relative to Wuhan 1 or Spike that also had the polymorphisms from the boxed haplotype in Fig 2. These viruses were used to infect cells after incubation in a dilution series of monoclonal antibodies known to neutralize Wuhan 1 and infectivity measured by luciferase activity. Light units were normalized to the maximum value per plate. Values represent three independent experiments done in triplicate. Error bars are the standard deviation between the experiments.

## Materials & methods

### Sample collection and RNA extraction

Collection and processing of samples were as previously described [2]. Twenty-four-hour composite samples were collected at the MO45 wastewater treatment facility and maintained at 4°C until they were delivered to the analysis lab, generally within 24 h of collection. Samples were then centrifuged at 3000× *g* for 10 min and followed by filtration through a 0.22 µM poly-ethersolfone membrane (Millipore, Burlington, MA, USA). Approximately 37.5 mL of waste-water was mixed with 12.5 mL solution containing 50% (*w/vol*) polyethylene glycol 8000 and 1.2 M NaCl, mixed, and incubated at 4°C for at least 1 h. Samples were then centrifuged at 12,000× *g* for 2 h at 4°C. Supernatant was decanted and RNA was extracted from the remaining pellet (usually not visible) with the QIAamp Viral RNA Mini Kit (Qiagen, Germantown, MD, USA) using the manufacturer's instructions. RNA was extracted in a final volume of 60 µL.

## MiSeq

Similar to our previous protocol [2], the primary RBD RT-PCR was performed using the Superscript IV One-Step RT-PCR System (Thermo Fisher Scientific,12594100). Primary RT-PCR amplification was performed as follows: 25˚C (2:00) + 50˚C (20:00) + 95˚C (2:00) + [95˚C (0:15) + 55˚C (0:30) + 72˚C (1:00)] × 25 cycles using the MiSeq primary PCR primers CTGCTTTACTAATGTCTATGCAGATTC and NCCTGATAAAGAACAGCAACCT. Secondary PCR (25 μL) was performed on RBD amplifications using 5 μL of the primary PCR as template with MiSeq nested gene specific primers containing 5′ adapter sequences (0.5 μM each) acactctttccctacacgacgctcttccgatctGTRATGAAGTCAGMCAAATYGC and gtgactggagttcagacgtgtgctcttccgatctATGTCAAGAATCTCAAGTGTCTG, dNTPs (100 μM each) (New England Biolabs, N0447L) and Q5 DNA polymerase (New England Biolabs, M0541S). Secondary PCR amplification was performed as follows: 95˚C (2:00) + [95˚C (0:15) + 55˚C (0:30) + 72˚C (1:00)] × 20 cycles.

For Omicron exclusion amplification, the primary RBD RT-PCR was performed using the MiSeq primary PCR primers ATTCTGTCCTATATAATTCCGCAT and CCCTGATAAA-GAACAGCAACCT (the first primer was changed to TATATAATTCCGCATCATTTTCCAC starting in May, 2022 to adapt to changing Omicron lineages) and secondary PCR used MiSeq nested gene specific primers containing 5′ adapter sequences (0.5 μM each) acactctttccctacac-gacgctcttccgatctGTGATGAAGTCAGACAAATCGC and gtgactggagttcagacgtgtgctcttccgatctATGTCAAGAATCTCAAGTGTCTG.

A tertiary PCR (50 μL) was performed to add adapter sequences required for Illumina cluster generation with forward and reverse primers (0.2 μM each), dNTPs (200 μM each) (New England Biolabs, N0447L) and Phusion High-Fidelity or (KAPA HiFi for CA samples) DNA Polymerase (1U) (New England Biolabs, M0530L). PCR amplification was performed as follows: 98˚C (3:00) + [98˚C (0:15) + 50˚C (0:30) + 72˚C (0:30)] × 7 cycles +72˚C (7:00). Amplified product (10 μl) from each PCR reaction is combined and thoroughly mixed to make a single pool. Pooled amplicons were purified by addition of Axygen AxyPrep MagPCR Cleanup beads (Axygen, MAG-PCR-CL-50) or in a 1.0 ratio to purify final amplicons. The final amplicon library pool was evaluated using the Agilent Fragment Analyzer automated electrophoresis system, quantified using the Qubit HS dsDNA assay (Invitrogen), and diluted according to Illumina's standard protocol. The Illumina MiSeq instrument was used to generate paired-end 300 base pair reads. Adapter sequences were trimmed from output sequences using Cutadapt. Raw sequences are available in NCBI's Sequence Read Archive under the bioproject PRJNA748354.

## Computational analysis

Sequencing reads were processed similar to previously described [2]. Briefly, BBTools (Bushnell B. – http://sourceforge.net/projects/bbmap/) were used to merge paired reads, which were dereplicated with a custom script (https://github.com/degregory/Programs/blob/main/derep.py). Dereplicated sequences from RBD amplicons were mapped to the reference sequence of SARS-CoV-2 (NC_045512.2) Spike ORF using Minimap2 [9]. Mapped amplicon sequences were then processed with SAM Refiner using the same spike sequence as a reference and the command line parameters "--Alpha 1.6 --foldab 0.6".

For Fig 1, SAM Refiner covariant deconvolution outputs were matched to defined variants to determine the relative abundance for each sample using a custom script (modified from https://github.com/istaves/covid-variant-counter). For Fig 2, the same outputs of SAM Refiner for MiSeq sequences were collected and were processed to determine core haplotypes of the cryptic lineage. First sequences that contained fewer than 4 polymorphisms relative to the

reference Wuhan I sequence or matched officially named variants were discarded. Remaining sequences were then processed to remove polymorphisms that never appeared in a sample at an abundance greater than .5%. In-frame deletions bypassed this removal. Condensed sequences that appear in at least two samples or had a summed abundance of at least 2% across all samples were passed on to further steps. All sequences were rendered into the figures using plotnine.

## Plasmids

The Δ19 spike clone with the D614G mutation, as previously described [10], was used as a starting template to generate the 22-10-04 spike and was digested using the restriction enzymes SacII and XhoI (New England Biolabs). The 12 missense mutations found in 22-10-04 were designed using A Plasmid Editor (ApE [11]), ordered as a gBlock (Integrated DNA Technologies), and inserted into the D614G Δ19 spike vector at the SacII and XhoI sites using the 5x In-Fusion Snap Assembly Master Mix (Takara Bio). The 22-10-04 clone was confirmed via Sanger Sequencing (Genomics Technology Core - University of Missouri). The vector for HIV-1-Gluc particles was previously described [12]. The GFP-N1 plasmid was originally provided by Clontech. All mAb plasmids were initially obtained as 4 μg maxipreps from Genscript, transformed into DH5α cells, and then isolated using the PureLink HiPure Plasmid Maxiprep Kit (Invitrogen). The plasmids for bamlanivimab included LY_CoV555_HC_pcDNA3.4 and LY_CoV555_LC_pcDNA3.4. The plasmids for etesevimab included CB6_HC_pcDNA3.4 and CB6_LC_pcDNA3.4. The plasmids for casirivimab included REGN10933_HC_pcDNA3.4 and REGN10933_LC_pcDNA3.4. The plasmids for imdevimab included REGN10987_HC_pcDNA3.4 and REGN10987_LC_pcDNA3.4.

## Cell culture

The 293FT/ACE2/TMPRSS2 cells, as previously described [10], and the 293FT cells (ATCC) were maintained in Dulbecco's Modified Eagle's Medium (DMEM; Sigma-Aldrich) supplemented with 7.5% fetal bovine serum (FBS; Gibco), 1% minimum essential medium (MEM) vitamins, 1% non-essential amino acids (NEAA), 4 mM L-Glutamine, and 1 mM sodium pyruvate (Sigma-Aldrich).

## Virus and antibody production

All transfections were done in 10 cm dishes with 293FT cells using polyethylenimine (PEI). For both HIV-1-Gluc pseudotyped particles, 9000 ng of provirus was co-transfected with 1000 ng of either the D614G Δ19 or the 22-10-04 spike expression vectors. For all mAb transfections (bamlanivimab, etesevimab, casirivimab, and imdevimab), 5000 ng of each light chain vector was co-transfected with 5000 ng of each respective heavy chain vector. Additionally, 300 ng of a GFP-N1 plasmid was co-transfected during both pseudoparticle and mAb productions to check for transfection efficiency, which was assessed using an Olympus IX70 Fluorescence Microscope to look for GFP production. At 3 days post-transfection, viral or mAb supernatant was collected and centrifuged at $3000 \times g$ for 5 minutes, and then the supernatant was separated from the pellet and collected and frozen at -80°C for a minimum of 4 hours before use in the neutralization assays.

## Neutralization and Gluc assays

During the viral neutralization assays, 96-well plates were prepared for mAb assessment. 50 μL of supplemented DMEM was added to wells 2-12. The first well of each plate received 100 μL

of only antibody supernatant, and then the subsequent wells were serially diluted by transferring supernatant from the previous well. At the tenth well of each row, 50 μL of the dilution was discarded, ensuring a consistent volume of 50 μL in all wells.

Next, 50 μL of viral supernatant was added to wells 1-11 so that wells 11 in all rows serve as positive controls containing only virus, media, and cells (added after incubation). In contrast, wells 12 were negative controls with only cells (added after incubation) and media. The plates were incubated at 37˚C with 5% $CO_2$ for at least 1 hour. Following incubation, ~28,000 of the 293FT/ACE2/TMPRSS2 cells were added to each well of the 96-well plates and then incubated for 2 days. All neutralizations were performed in triplicate on each plate.

Two days post-infection, 20 μL of supernatant from all wells was collected and placed into a black 96-well plate. Then, 50 μl of 25 mM coelenterazine (CTZ; Nanolight Technology) suspended in phosphate-buffered saline (PBS; Cytiva) was added to all wells and placed into a PerkinElmer EnSpire 2300 Multilabel Reader to measure *Gaussia* luciferase activity. Infectivity was measured as relative light units (RLU).

## Acknowledgments

We would like to acknowledge the University of Missouri Bioinformatics and Analytics Core for their services with MiSeq sequencing.

## Author Contributions

**Conceptualization:** Devon A. Gregory, Chung-Ho Lin, Elizabeth Semkiw, Melissa Reynolds, Jeff Wenzel, Marc C. Johnson.

**Formal analysis:** Devon A. Gregory, Marc C. Johnson.

**Funding acquisition:** Chung-Ho Lin, Melissa Reynolds, Jeff Wenzel, Marc C. Johnson.

**Investigation:** Devon A. Gregory, Clayton Rushford, Torin Hunter, Chung-Ho Lin, Christie Darby, Nicole Niehues, Elizabeth Semkiw, Melissa Reynolds, Jeff Wenzel, Marc C. Johnson.

**Writing – original draft:** Devon A. Gregory, Torin Hunter, Melissa Reynolds, Jeff Wenzel, Marc C. Johnson.

**Writing – review & editing:** Devon A. Gregory, Torin Hunter, Melissa Reynolds, Jeff Wenzel, Marc C. Johnson.

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
