## [Decision Letter · Decision Letter 0]

16 Oct 2023

Dear Dr. Johnson,

Thank you very much for submitting your manuscript "Continued selection on cryptic SARS-CoV-2 observed in Missouri wastewater" for consideration at PLOS Pathogens. As with all papers reviewed by the journal, your manuscript was reviewed by members of the editorial board and by several independent reviewers. The reviewers appreciated the attention to an important topic. Based on the reviews, we are likely to accept this manuscript for publication, providing that you modify the manuscript according to the review recommendations.

We note that Reviewer 1 raised several important concerns that should be addressed in the manuscript text.

Sincerely,

Thomas E. Morrison

Academic Editor

PLOS Pathogens

Alexander Gorbalenya

Section Editor

PLOS Pathogens

Kasturi Haldar

Editor-in-Chief

PLOS Pathogens

orcid.org/0000-0001-5065-158X

Michael Malim

Editor-in-Chief

PLOS Pathogens

orcid.org/0000-0002-7699-2064

Reviewer Comments (if any, and for reference):

Reviewer's Responses to Questions

**Part I - Summary**

Reviewer #1: This is a very brief research article from a leading group in wastewater surveillance of SARS-CoV-2. The authors report continued evolution of the Spike gene in a single cryptic lineage they identified in Missouri wastewater.

The authors are only able to assess changes in the RBD, so the data do not provide a comprehensive view of the cryptic lineage. However, there are considerable technical challenges in obtaining whole viral genome sequences from sewage, so one can argue the focus on the RBD is sensible. In any case, the authors observe missense changes within the Spike RBD that are similar to those observed in current circulating Omicron lineages.

The main concern I have is whether this brief, observational study is appropriate for PLOS Pathogens. Although the work will no doubt be of interest to many people in the virology field, and beyond, the observation of convergent evolution in the Spike gene is no longer novel or original, but rather has become commonplace and is to-be-expected. Moreover, the manuscript does not include any studies of how these amino acid changes impact viral entry, antibody escape, et cetera. Most reports evaluating novel variants include functional, "wet lab" analyses, at least using pseudotyped viruses, to evaluate the phenotypic outcomes of the observed nucleotide changes. That is lacking here. Though again, the authors only have the RBD, so there'd be a question of what sort of Spike the authors should place these changes into for any proposed laboratory studies.

Otherwise, the authors methodology and approach are straightforward, and the manuscript is well written. The authors' take-home point /argument that surveillance of cryptic lineages is valuable / worthwhile in providing a signal of changes yet-to-come in circulating variants is valid.

Reviewer #2: In this short report Gregory and coauthors describe the long-term and ongoing detection of a ‘cryptic’ wastewater lineage – what is presumed to be a undetected chronic infection, perhaps in an immunocompromised host. The authors draw parallels between mutations appearing in this lineage and the same sites of substitutions appearing at later dates in Omicron or Omicron sublineages. This topic is of real interest and the authors tell a compelling story. I would recommend the paper is accepted for publication, I have some very minor suggestions (including one I don’t think needs directly addressing by the authors) below.

Reviewer #3: This is a short report on a very interesting highly mutated SARS-CoV-2 variant found in a Missouri watershed.

It should be of substantial interest as it chronicles the extensive long-term evolution of this sample.

I only have a few minor comments:

Line 53: it is never introduced what "MO45" means.

The manuscript lacks a data availability statement on where the raw sequencing data can be found.

**Part II – Major Issues: Key Experiments Required for Acceptance**

Reviewer #1: (No Response)

Reviewer #2: No major issues

Reviewer #3: (No Response)

**Part III – Minor Issues: Editorial and Data Presentation Modifications**

Reviewer #1: (No Response)

Reviewer #2: • Dates on the X axis of Figure 1 have been cut off.

• One slight worry I have about the method used – specifically the Omicron exclusion primers – if, as the authors suggest, the cryptic lineages show convergent evolution with Omicron – isn’t there a concern that the sites covered by the primers could mutate in these cryptic lineages and result in a lack of sensitivity? For example site 368 and sites 371-373. I don’t think this is what is happening in this case (as the lineage is clearly still detected) but it might be worth considering outside the bounds of this study.

Reviewer #3: (No Response)

PLOS authors have the option to publish the peer review history of their article (what does this mean?). If published, this will include your full peer review and any attached files.

Reviewer #1: No

Reviewer #2: **Yes: **Thomas P. Peacock

Reviewer #3: No

Figure Files:

Data Requirements:

Reproducibility:

References:

---

## [Editor Report · Decision Letter 1]

18 Dec 2023

Dear Dr. Johnson,

We are pleased to inform you that your manuscript 'Continued selection on cryptic SARS-CoV-2 observed in Missouri wastewater' has been provisionally accepted for publication in PLOS Pathogens.

Best regards,

Thomas E. Morrison

Academic Editor

PLOS Pathogens

Alexander Gorbalenya

Section Editor

PLOS Pathogens

Kasturi Haldar

Editor-in-Chief

PLOS Pathogens

orcid.org/0000-0001-5065-158X

Michael Malim

Editor-in-Chief

PLOS Pathogens

orcid.org/0000-0002-7699-2064
---

## [Editor Report · Acceptance letter]

21 Dec 2023

Dear Dr. Johnson,

We are delighted to inform you that your manuscript, "Continued selection on cryptic SARS-CoV-2 observed in Missouri wastewater," has been formally accepted for publication in PLOS Pathogens.

Best regards,

Michael Malim

Editor-in-Chief

PLOS Pathogens

orcid.org/0000-0002-7699-2064